# Primary Healthcare Professionals’ Knowledge and Attitudes Towards Meningococcal, Rotavirus, and HPV Vaccines in Children and Adolescents

**DOI:** 10.3390/healthcare13070811

**Published:** 2025-04-03

**Authors:** Eren Yıldız, Rukiye Ünsal Saç, Hilmi Onur Kabukçu, Fethiye Yıldız, Funda Çatan İnan, Medine Ayşin Taşar

**Affiliations:** 1Department of Pediatrics, Kastamonu University Faculty of Medicine, Kastamonu 37150, Turkey; 2Department of Pediatrics, University of Health Sciences, Ankara Training and Education Hospital, Ankara 06230, Turkey; 3Department of Biostatistics and Medical Informatics, Faculty of Medicine, Bilecik Seyh Edebali University, Bilecik 11200, Turkey

**Keywords:** vaccine, physician, attitude, knowledge, healthcare professionals

## Abstract

**Aim:** This study aimed to evaluate the knowledge and attitudes of family physicians and family health personnel who are responsible for childhood vaccination services in primary care regarding meningococcal, rotavirus, and human papillomavirus vaccines. **Methods:** This cross-sectional study was conducted between October 2021 and January 2022. A total of 700 healthcare professionals from all geographical regions in Turkey were included in the study. The participants filled out an online questionnaire consisting of 39 questions created with Google Forms. **Results:** Of the participants, 340 (48.6%) are family physicians, and 360 (51.4%) are family health personnel. Most participants are from the Marmara region, Turkey’s most densely populated region. The most recommended vaccine among the participants was rotavirus (84.3%), while the least recommended vaccine was human papilloma virus (47.6%). The number of family physicians recommending meningococcal and human papilloma virus vaccines was significantly higher than that of family health personnel (*p* < 0.001). Furthermore, there was a statistically significant correlation between seeing a patient with these viruses in one’s professional life or considering the severity of these three diseases to be severe and recommending these vaccines (*p* < 0.001 for both). However, lack of knowledge about vaccines and doses, the high cost of vaccines, and concerns about side effects were among the reasons for not recommending vaccines. **Conclusions:** Healthcare professionals involved in childhood immunization should be trained to increase their knowledge and awareness on this issue. The training plan and curriculum should take into account the issues raised in our research, such as age, occupation, region of residence, and professional experience. In general, knowledge about the efficacy and safety of vaccines will help healthcare professionals develop their confidence in vaccines and willingness to recommend childhood vaccines to others.

## 1. Introduction

Vaccination is the most effective intervention in protecting child and adult health and reducing mortality and morbidity due to infectious diseases [1]. In Turkey, an immunization program has been successfully carried out for a long time within the framework of the Extended Immunization Program. While childhood vaccination rates in Turkey were 78–82% in 2002, this rate increased to 97–99% in 2019 [2]. Per the national vaccination calendar applied in Turkey, hepatitis B (HepB); tuberculosis (BCG); thirteen-valent conjugated pneumococcal (PCV13); five-valiant combination inclusive of diphtheria, tetanus, and acellular pertussis (DTaP); live attenuated oral polio (OPV); measles–mumps–rubella (MMR) combination; varicella (VAR); and hepatitis A (HepA) vaccines are routinely administered to children [3]. In addition, rotavirus, meningococcal, and human papillomavirus (HPV) vaccines, which are not included in the national vaccine scheme, are administered in health centers with doctors’ recommendation and following a fee paid by the family.

Rotavirus is the most common cause of acute gastroenteritis (AGE) in underdeveloped and developing countries [4]. It is the most common cause of AGE, especially in children under two, and causes AGE-related hospitalizations [5]. In a study conducted in Turkey, rotavirus positivity was found in 8.6% of pediatric patients diagnosed with AGE [5]. Vaccination is the main method to prevent mortality and morbidity caused by this infection. Rotavirus vaccines (RVs) are easy to use and administered orally. Vaccination has been shown to reduce rotavirus-related gastroenteritis and hospitalizations [6].

Thirteen subtypes of sexually transmitted HPVs are responsible for all cervical cancers (57,000 cases per year); some oropharyngeal cancers (120,000 cases per year); anogenital warts; recurrent respiratory papillomatosis; and anal, vaginal, vulvar, and penile cancers [7]. The human papilloma virus vaccine (HPVV) is the most effective method of preventing HPV infection [8]. The HPVV achieves permanent and effective immunity and prevents chronic infection that may progress to carcinoma [9].

Meningococcal disease caused by *Neisseria meningitidis* occurs in a wide spectrum, from asymptomatic carriage to life-threatening invasive meningococcal disease (IMD) [10]. Nearly all IMD is caused by six meningococcal serogroups: A, B, C, W, X, and Y (MenA, MenB, MenC, MenW, MenX, and MenY, respectively) [11]. Except for serogroup X, vaccines are available for all major disease-causing serogroups of *N. meningitidis* (A, B, C, W, and Y) [12]. In Turkey, three licensed vaccines combine the A, C, W, and Y strains. However, only one licensed monovalent vaccine for serogroup B is present [13].

Childhood vaccination practices in Turkey are primarily carried out in family health centers, where family physicians and family health personnel (nurses or midwives) work in collaboration to deliver immunization services. These professionals play a pivotal role not only in administering vaccines but also in providing education and guidance to families. Family physicians and other frontline healthcare professionals serve as a key source of vaccine-related information and are instrumental in shaping public attitudes. Indeed, provider recommendations have consistently been shown to be among the strongest predictors of vaccine acceptance [14,15]. Healthcare providers who are well informed and hold positive attitudes toward immunization are more likely to recommend vaccines confidently, which, in turn, contributes to higher uptake. Conversely, when providers lack sufficient knowledge or confidence, they may hesitate to recommend vaccines, potentially reinforcing vaccine hesitancy among the public [16,17].

In Turkey, rotavirus, meningococcal, and human papillomavirus (HPV) vaccines—although important for childhood immunization—are not included in the national vaccination schedule, and families often rely on physicians’ recommendations when deciding whether to administer them. A recent Turkish survey found that family physicians with greater knowledge of these non-program vaccines were significantly more likely to recommend them, while limited knowledge was associated with lower recommendation and uptake rates [18]. Similarly, international studies have emphasized that healthcare professionals are among the most trusted sources of vaccine information and that their personal beliefs and attitudes can directly influence clinical recommendations and parental acceptance of childhood vaccines [19]. Despite the recognized influence of primary care providers on immunization behavior, there appears to be limited research examining how primary care teams—particularly the collaborative efforts of family physicians and family health personnel—support the delivery of childhood vaccines that fall outside the national immunization program. Addressing this gap, the present study aimed to assess the knowledge and attitudes of healthcare professionals working in family health centers regarding non-routine childhood vaccines.

## 2. Methods

This cross-sectional study was conducted between October 2021 and January 2022. Ethical approval (dated 22 September 2021 and numbered 2020-KAEK-143-116) was obtained from the Kastamonu University Clinical Research Ethics Committee.

There are approximately 30,000 family physicians in Turkey. Each family physician works at a family health center by forming a family medicine unit with family health personnel (a nurse or midwife). If the job definition of the family health center is to be briefly described, it is the union of family medicine units consisting of family physicians and family health personnel who provide preventive and curative services [20].

In our study, participants were recruited from all geographical regions in Turkey. Participants were not recruited from all family health centers across Turkey. Instead, voluntary response sampling was used to include professionals from randomly selected centers located in various geographical regions. Invitations were shared via professional email groups and social media platforms. Electronic informed consent was obtained from all participants before starting the questionnaire. The study population comprised 700 healthcare professionals (family physicians or family health personnel) working in primary family health centers.

An online questionnaire created with Google Forms was presented to the participants. Questionnaire items were prepared using a 5-point Likert-type frequency scale, with multiple-choice, closed-ended and open-ended questions, and there was a total of 39 questions.

To estimate the required sample size, G*Power analysis was used, taking α = 0.05 and power = 0.95. In addition, the effect size of our pilot study (Npilot = 50) was calculated as small to medium. Therefore, a small- to medium-sized effect was assumed for the power analysis. For the chi-square test, the sample was determined to be at least n = 522. Finally, we aimed to include 700 healthcare professionals.

The questionnaire was developed specifically for this study based on a review of the relevant literature in the field. Since the instrument was not a standardized or validated scale, no formal reliability or validity testing (e.g., Cronbach’s alpha) was applied. The questionnaire included several open-ended questions designed to explore participants’ reasoning for recommending or not recommending certain vaccines.

The participants were contacted only once. No follow-ups or reminders were sent. Participation was completely voluntary. Since the questionnaire was not a validated scale with a scoring system, knowledge and attitudes were not classified into levels such as low, moderate, or high. Instead, the participants’ attitudes were evaluated based on whether they recommended each vaccine.

All statistical analyses were performed using SPSS 23.00 (SPSS Inc., Chicago, IL, USA). Descriptive statistics were used to summarize categorical variables as frequencies and percentages. In addition, Pearson’s chi-square test was conducted to determine the relationship in proportions of categorical variables between groups. A *p*-value of <0.05 was considered significant.

## 3. Results

A total of 700 healthcare professionals participated in this research. Of the participants, 340 (48.6%) were family physicians, and 360 (51.4%) were family health personnel (Table 1). The participants were between the ages of 23 and 64, and the mean age was 38.8 ± 7.9 years. Furthermore, the mean age was 38.5 ± 7.5 years for females and 39.5 ± 8.8 years for males. The mean working years as a healthcare professional was 15.6 ± 8.5 years. Most family physicians (83.5%) were practitioners without residency training. In addition, most of the participants (23.7%) were from the Marmara region, Turkey’s most densely populated area.

The distribution of the participants’ attitudes towards vaccines not in the routine schedule—RV, meningococcal vaccine (MV), and HPVV—are shown in Table 2 Among the participants, the most recommended vaccine was the RV (84.3%), and the least recommended vaccine was the HPVV (47.6%). In addition, 58.4% of male and 42.2% of female healthcare professionals recommended HPVV treatment (χ^2^ = 16.33, *p* < 0.001).

A significant difference was found between the ages of the healthcare professionals and the recommendation of vaccines that are not in the national vaccine scheme. Healthcare professionals aged 41–65 recommended vaccines that are not included in the national vaccination calendar to their patients less frequently than healthcare professionals in other age groups (χ^2^ = 6.19, *p* = 0.04). Again, the percentage of healthcare professionals in this age group recommending the RV was 79.5% lower than the other age group healthcare professionals (χ^2^ = 8.02, *p* = 0.01).

The number of family physicians recommending the meningococcal vaccine and HPVV (81.8% and 58.5%, respectively) was statistically higher than the number of family health personnel (70.3% and 37.2%, respectively) (*p* < 0.001 for both). The rate of healthcare professionals who least recommended the MV (55.6%) was in the Southeastern Anatolia region (*p* = 0.01). The percentage of recommending the MV was the highest in the group of healthcare professionals whose time spent in the profession was >21 years (*p* = 0.02).

A significant difference was found between seeing rotavirus, meningococcal, and HPV-related diseases in professional life and recommending vaccines to prevent these diseases (*p* < 0.001 for each) (Table 3).

A statistically significant difference was found between the opinions of healthcare professionals about the severity of rotavirus, meningococcal, and HPV-related diseases and their recommendations for vaccination to prevent these diseases (each, *p* < 0.001) (Table 4). Healthcare professionals who thought these diseases were fatal and serious recommended vaccines at a higher rate for each disease.

Most participants (n = 511, 73%) thought there was no relationship between rotavirus vaccine recommendation and its easy administration (orally administered). When the participants were asked which of the meningococcal vaccines they recommended, they most frequently recommended only the MEN ACWY vaccine (n = 322, 72.2%). Only 3.3% (n = 15) recommended MEN B vaccines together with MEN ACWY. Of the participants, 92% (n = 644) knew that HPV causes cervical cancer.

Of the participants, 63.7% recommended the RV (n = 376) and 59.7% the MV (n = 317) for all followed patients. In addition, they recommended the HPVV to their patients (n = 214, 64.2) who wanted to be vaccinated (Table 5).

When the reasons for not recommending vaccines were queried, the most common reason for not recommending the MV (61.5%) and HPVV (72.5%) was a lack of knowledge about the vaccine and dose. The most common reason for not recommending the RV (50%) was its high price. However, concerns about the side effects of vaccines were among the reasons why the vaccine was not recommended.

## 4. Discussion

The most recommended vaccine was the RV, and the least recommended vaccine was for HPV. Being a woman, being between the ages of 41 and 65, being a family health worker, participating from the Southeastern Anatolia region, not knowing about the vaccine and its dose, thinking that the vaccine is expensive, and worrying about its side effects are some of the negative factors affecting the attitude of recommending the vaccine. While most of those recommending the meningococcal vaccine recommended the Men ACWY vaccine alone, the frequency of those recommending the MenB and MenACWY vaccines was 3%.

In the study of Çataklı et al., in which they examined the attitudes of family physicians and pediatricians towards the RV, HPV, MenACWY-MCV4, and adolescent/adult pertussis vaccines, which are not included in the national immunization program, the RV was recommended at 60.5%, MCV4 at 52.6%, and HPVV at 45.6% [3]. In our study, the RV was recommended with a frequency of 84.2%, the MV with a frequency of 75.8%, and the HPVV with a frequency of 47.5%. Although the recommended vaccine rates in Çataklı et al.’s study were lower than ours, the RV was the most frequently recommended vaccine in both studies. The reason the RV is being recommended more and more among healthcare professionals is that the importance given to these vaccines in Turkey is increasing every year and is still the most common cause of AGE [21].

Our study found that healthcare professionals who thought that rotavirus, meningococcal, and HPV-related diseases were fatal and serious recommended vaccines at a higher rate for each disease. Like our study, Topuridze M et al. reported that healthcare workers with a low perception of disease severity were less likely to recommend vaccination [22]. However, a repeat cross-sectional study examining pediatric healthcare professionals’ experience with the RV in Sweden showed the general application of the RV being perceived as successful, positive, and easier than expected among pediatric healthcare professionals [23].

A systematic review reported that knowledge of vaccines and their efficacy and safety helped healthcare professionals develop their confidence and willingness to recommend vaccines to others [14]. The main reasons why vaccines were not recommended in our study were not knowing the vaccine and its dose, the vaccine being too expensive, and worrying about its side effects, respectively. Similar to these findings, many studies state that insufficient knowledge about vaccines, their expensiveness, and concerns about their side effects lead to not recommending the vaccine or vaccine hesitation [3,24,25].

In our study, only 3.3% of healthcare professionals recommended that the MenACWY and MenB vaccines be administered together. Vaccination is the most effective way to prevent meningococcal disease. For complete protection, it is recommended to give both the MenACWY and MenB vaccines because serogroup B dominates some years, and other serogroups dominate other years [26,27]. This study showed that healthcare professionals’ knowledge about the necessity of both types of meningococcal vaccines for full protection is insufficient. Previous studies reported that the reasons for meningococcal vaccination were being recommended by healthcare professionals, sufficient knowledge about the vaccine, positive attitude, perceived risk, and easy accessibility [28,29,30]. Reasons for refusal of vaccination have been reported as a lack of sufficient information and low risk perception [28,31]. In our study, the results for physicians recommending the MV more than other healthcare professionals, regional differences, and recommending more vaccinations for those exposed to meningococcal disease and less vaccination for those who were undecided about the severity of the disease were similar to the results of the above publications.

In our study, 58.5% of the doctors and only 37.2% of the family health personnel routinely recommended the HPV vaccine. Doctors and those who have seen the HPV disease in their professional life recommend the HPV vaccine more than those who are undecided about the severity of the disease, which can be explained by the level of knowledge and awareness. A study conducted among healthcare professionals reported that the most common reason for not getting the HPV vaccine was a lack of awareness [32]. A study whose sample consisted only of nurses reported that the main perception barriers in HPV vaccine advocacy were insufficient education and knowledge about HPV and not feeling comfortable discussing the HPV vaccine with patients [33].

Among healthcare professionals, knowledge and attitudes toward vaccination are important in the parents’ decisions to vaccinate their children [23]. However, it is also very important for healthcare professionals to be aware of their impressive role and to display reassuring attitudes in implementing new preventive health services such as new vaccines [14]. In addition, effective parent communication can shape parents’ attitudes toward vaccination and make it easier for healthcare professionals to understand the family’s situation and needs [34]. Additionally, vaccination recommendations were more common when healthcare professionals felt comfortable in explaining the risks and benefits [14]. In addition to attitudes, other psychological and social factors play important roles in shaping healthcare professionals’ vaccination behavior. According to the Theory of Planned Behavior, behavioral intention is influenced not only by individual attitudes but also by subjective norms (perceptions of what others expect) and perceived behavioral control (confidence in one’s ability to perform the behavior) [35]. These constructs may significantly affect whether a professional recommends a vaccine. Future studies should consider these broader determinants to better understand the complex nature of vaccine recommendation behavior among primary healthcare professionals.

Healthcare professionals play a key role in the increase in the demand for these three vaccines, which are not in the national vaccine program but are allowed to be used. Therefore, this research is important in revealing some effective factors for healthcare professionals to recommend the vaccine. Effective targeted interventions can be designed by considering these factors and other research results.

Using an online survey to better understand participants’ awareness is a key strength of our study. However, there are also limitations associated with this study. Before participating in the survey, the respondents were informed of the survey being about “people’s attitudes towards certain vaccines.” Therefore, there may be a selection bias for participants with a positive perception of the vaccine. Also, because the research was conducted using an online panel, the generalization of the findings may be limited due to possible selection bias. Additionally, since the questionnaire used in this study was developed specifically for the research and was not a validated or standardized scale, its psychometric properties (e.g., internal consistency and construct validity) were not formally assessed. Moreover, as some questions were open ended, responses may be subject to interpretation bias. These factors should be considered when interpreting the results. Finally, the lack of an interviewer or pre-test that would enable the participants to understand the questions may also have affected the answers and findings of the study.

## 5. Conclusions

Training should be given to increase the knowledge and awareness of healthcare professionals on this issue. The training plan and curriculum should be prepared in such a way as to consider the issues raised in our research, such as age, occupation, region of residence, and professional experience. Knowledge of specific vaccines, their efficacy, and safety will help healthcare professionals develop their confidence and willingness to recommend vaccines. However, as with other new vaccines, these three vaccines run the risk of being low in practice unless publicly funded.

This study reveals a significant deficiency in the recommendation practices of non-mandatory childhood vaccines among primary healthcare professionals in Turkey. Although they occupy a critical position in the delivery of preventive healthcare, a considerable proportion of professionals do not routinely recommend vaccines such as rotavirus, meningococcal, and HPV vaccines. The predominant barriers identified include inadequate knowledge—reported by 72.5% for HPV and 61.5% for meningococcal vaccines—along with concerns related to vaccine cost and potential side effects. These findings highlight the urgent need for comprehensive, evidence-based educational interventions and stronger national policy frameworks to support and empower healthcare providers. Enhancing their knowledge base and confidence is essential to not only to improve vaccine recommendation rates but also to address vaccine hesitancy more effectively and contribute to broader public health goals.

## Figures and Tables

**Table 1 healthcare-13-00811-t001:** Distribution of descriptive characteristics of the participants.

	n	%
Sex		
Female	469	67
Male	231	33
Age (years)		
21–30	114	16.3
31–40	308	44.0
41–65	278	39.7
Place of work		
Urban	310	44.3
Rural	390	55.7
Occupation		
Family physician	340	48.6
Family health staff	360	51.4
Type of family physician (n = 340)		
Practitioner without residency training	284	83.5
Specialist physician	56	16.5
Education level of family health staff		
Postgraduate	28	7.7
Undergraduate	229	63.6
Associate degree	69	19.2
High school graduate	34	9.5
Geographical region		
Marmara	166	23.7
Central Anatolia	157	22.4
Black Sea	117	16.7
Mediterranean	80	11.4
Aegean	72	10.3
Southeastern Anatolia	66	9.4
Eastern Anatolia	42	6.0
Duration of working as a healthcare professional (years)		
1–10	244	34.9
11–20	252	36.0
>20	204	29.1
Duration of working in primary healthcare services (years)		
1–10	408	58.3
11–20	190	27.1
>20	102	14.6

**Table 2 healthcare-13-00811-t002:** Relationship between participants’ characteristics and their attitudes towards non-routine vaccines: RV, MV, and HPV.

	Non-Routine		RV		MV		HPV	
	R	NR	R	NR	R	NR	R	NR
Sex								
Female	359 (76.5)	110 (23.5)	398 (84.9)	71 (15.1)	349 (74.4)	120 (25.6)	198 (42.2)	271 (57.8)
Male	167 (72.3)	64 (27.7)	192 (83.1)	39 (16.9)	182 (78.8)	49 (21.2)	135 (58.4)	96 (41.6)
	*χ*^2^ = 1.49	*p* = 0.22	χ^2^ = 0.35	*p* = 0.55	χ^2^ = 1.61	*p* = 0.20	χ^2^ = 16.33	*p* < 0.001
Age								
21–30	90 (78.9)	24 (21.1)	99 (86.8)	15 (13.2)	83 (72.8)	31 (27.2)	49 (43.0)	65 (57.0)
31–40	241 (78.2)	67 (21.8)	270 (87.7)	38 (12.3)	235 (76.3)	73 (23.7)	155 (50.3)	153 (49.7)
41–65	195 (70.1)	83 (29.9)	221 (79.5)	57 (20.5)	213 (76.6)	65 (23.4)	129 (46.4)	149 (53.6)
	χ^2^ = 6.19	*p* = 0.04	χ^2^ = 8.02	*p* = 0.01	χ^2^ = 0.70	*p* = 0.70	χ^2^ = 2.05	*p* = 0.35
Place of work								
Urban	243 (78.4)	67 (21.6)	264 (85.2)	46 (14.8)	246 (79.4)	64 (20.6)	143 (46.1)	167 (53.9)
Rural	283 (72.6)	107 (27.4)	326 (83.6)	64 (16.4)	285 (73.1)	105 (26.9)	190 (48.7)	200 (51.3)
	χ^2^ = 3.13	*p* = 0.07	χ^2^ = 0.32	*p* = 0.57	χ^2^ = 3.17	*p* = 0.05	χ^2^ = 0.46	*p* = 0.49
Occupation								
Family physician	260 (76.5)	80 (23.5)	287 (84.4)	53 (15.6)	278 (81.8)	62 (18.2)	199 (58.5)	141 (41.5)
Family health staff	266 (73.9)	94 (26.1)	303 (84.2)	57 (15.8)	253 (70.3)	107 (29.7)	134 (37.2)	226 (62.8)
	χ^2^ = 0.62	*p* = 0.43	χ^2^ = 0.00	*p* = 0.92	χ^2^ = 12.59	*p* < 0.001	χ^2^ = 31.82	*p* < 0.001
Type of family physician								
Untrained practitioner	216 (76.1)	68 (23.9)	235 (82.7)	49 (17.3)	228 (80.3)	56 (19.7)	168 (59.2)	116 (40.8)
Specialist physician	44 (78.6)	12 (21.4)	52 (92.9)	4 (7.1)	50 (89.3)	6 (10.7)	31 (55.4)	25 (44.6)
	χ^2^ = 0.16	*p* = 0.68	χ^2^ = 3.63	*p* = 0.05	χ^2^ = 2.54	*p* = 0.11	χ^2^ = 0.27	*p* = 0.59
Education level of family health staff								
Postgraduate	25 (89.3)	3 (10.7)	26 (92.9)	2 (7.1)	23 (82.1)	5 (17.9)	11 (39.3)	17 (60.7)
Undergraduate	163 (71.2)	66 (28.8)	191 (83.4)	38 (16.6)	159 (69.4)	70 (30.6)	85 (37.1)	144 (62.9)
Associate degree	52 (75.4)	17 (24.6)	56 (81.2)	13 (18.8)	49 (71.0)	20 (29.0)	26 (37.7)	43 (62.3)
High school graduate	26 (76.5)	8 (23.5)	30 (88.2)	4 (11.8)	22 (64.7)	12 (35.3)	12 (35.3)	22 (64.7)
	χ^2^ = 4.50	*p* = 0.21	*χ*^2^ = 2.57	*p* = 0.46	*χ*^2^ = 2.48	*p* = 0.47	χ^2^ = 0.11	*p* = 0.99
Geographical region								
Mediterranean	60 (75.0)	20 (25.0)	67 (83.8)	13 (16.3)	63 (78.8)	17 (21.3)	43 (53.8)	37 (46.2)
Eastern Anatolia	30 (71.4)	12 (28.6)	35 (83.3)	7 (16.7)	32 (76.2)	10 (23.8)	19 (45.2)	23 (54.8)
Aegean	55 (76.4)	17 (23.6)	61 (84.7)	11 (15.3)	57 (79.2)	15 (20.8)	38 (52.8)	34 (47.2)
Southeastern Anatolia	44 (66.7)	22 (33.3)	53 (80.3)	13 (19.7)	38 (57.6)	28 (42.4)	38 (57.6)	28 (42.4)
Central Anatolia	111 (70.7)	46 (29.3)	130 (82.8)	27 (17.2)	115 (73.2)	42 (26.8)	65 (41.4)	92 (58.6)
Black Sea	92 (78.6)	25 (21.4)	100 (85.5)	17 (14.5)	94 (80.3)	23 (19.7)	51 (43.6)	66 (56.4)
Marmara	134 (80.7)	32 (19.3)	144 (86.7)	22 (13.3)	132 (79.5)	34 (20.5)	79 (47.6)	87 (52.4)
	χ^2^ = 8.09	*p* = 0.23	χ^2^ = 1.99	*p* = 0.92	χ^2^ = 15.92	*p* = 0.01	χ^2^ = 7.88	*p* = 0.24
Duration of working as a healthcare professional (years)								
1–10	187 (76.6)	57 (23.4)	212 (86.9)	32 (13.1)	185 (75.8)	59 (24.2)	127 (52.0)	117 (48.0)
11–20	193 (76.6)	59 (23.4)	215 (35.3)	37 (14.7)	179 (71.0)	73 (29.0)	105 (41.7)	147 (58.3)
>21	146 (71.6)	58 (28.4)	163 (79.9)	41 (20.1)	167 (81.9)	37 (18.1)	101 (49.5)	103 (50.5)
	χ^2^ = 1.96	*p* = 0.37	χ^2^ = 4.40	*p* = 0.11	χ^2^ = 7.22	*p* = 0.02	χ^2^ = 5.79	*p* = 0.05
Duration of working in primary healthcare services (years)								
1–10	310 (76.0)	98 (24.0)	348 (85.3)	60 (14.7)	308 (75.5)	100 (24.5)	199 (48.8)	209 (51.2)
11–20	143 (75.3)	47 (24.7)	160 (84.2)	30 (11.8)	141 (74.2)	49 (25.8)	89 (46.8)	101 (53.2)
>21	73 (71.6)	29 (28.4)	82 (80.4)	20 (19.6)	82 (80.4)	20 (19.6)	45 (44.1)	57 (55.9)
	χ^2^ = 0.85	*p* = 0.65	χ^2^ = 1.48	*p* = 0.47	χ^2^ = 1.45	*p* = 0.48	χ^2^ = 0.76	*p* = 0.68

HPV: human papilloma virus vaccine, MV: meningococcal vaccine, NR: not recommending, R: recommending, and RV: rotavirus vaccine.

**Table 3 healthcare-13-00811-t003:** The relationship between seeing the disease caused by this virus in professional life and the attitude of recommending vaccination.

	Recommended	Not Recommended	X^2^	*p*
Have you ever cared for a patient with rotavirus gastroenteritis in your career?				
Yes	392 (88.1)	53 (11.9)	13.34	<0.001
No	198 (77.6)	57 (22.4)		
Have you ever cared for a patient with meningococcal disease in your career?				
Yes	210 (86.4)	33 (13.6)	22.67	<0.001
No	321 (70.2)	136 (29.8)		
Have you ever cared for a patient with an HPV-related disease in your career?				
Yes	213 (57.0)	161 (43.0)	28.33	<0.001
No	120 (36.8)	206 (63.2)		

HPV: human papilloma virus.

**Table 4 healthcare-13-00811-t004:** The relationship between the severity of the disease and the attitude to recommend vaccination.

	Recommended	Not Recommended	X^2^	*p*
What do you think about the severity of rotavirus?				
A simple disease	14 (41.2)	20 (58.8)		
Serious	379 (88.8)	48 (11.2)	108.29	<0.001
Deadly	161 (93.1)	12 (6.9)		
I’m indecisive	36 (54.5)	30 (45.5)		
What do you think about the severity of MV?				
A simple disease	0 (0)	0 (0)		
Serious	108 (64.7)	59 (35.3)	98.64	<0.001
Deadly	419 (83.5)	83 (16.5)		
I’m indecisive	4 (12.9)	27 (87.1)		
What do you think about the severity of HPV?				
A simple disease	15 (48.4)	16 (51.6)		
Serious	211 (50.5)	207 (49.5)	48.03	<0.001
Deadly	96 (57.5)	71 (42.5)		
I’m indecisive	11 (13.1)	73 (86.9)		

HPV: human papilloma virus vaccine; MV: meningococcal vaccine.

**Table 5 healthcare-13-00811-t005:** Type of patients and vaccine recommendation rates.

	Rotavirus	MV	HPV
	590	531	333
	n (%)	n (%)	n (%)
Type of patients and vaccine recommendation rates			
All patients under follow-up	376 (63.7)	317 (59.7)	81 (24.3)
Patients who have a chronic illness	54 (9.1)	87 (16.4)	29 (8.7)
Affluent patients	150 (25.4)	130 (24.5)	78 (23.4)
Willingness to vaccinate	237 (40.1)	230 (43.3)	214 (64.2)
Other	6 (1)	8 (1.5)	18 (5.4)

HPV: human papilloma virus vaccine; MV: meningococcal vaccine.

## Data Availability

Derived data supporting the findings of this study are available from the corresponding author (E.Y.) on request.

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
