# Peer review of "Primary Healthcare Professionals’ Knowledge and Attitudes Towards Meningococcal, Rotavirus, and HPV Vaccines in Children and Adolescents"

_healthcare, 2025, doi:10.3390/healthcare13070811_

Round 1
Reviewer 1 Report
Comments and Suggestions for Authors
The title “Evaluation of the knowledge and attitudes of primary care health professionals on Meningococcal, Rotavirus, and Human Papillomavirus vaccines” in this manuscript addresses an important area and will provide readers with valuable information. However, the title can be modified to convey the extent of the work that was conducted. Evaluation in this present context does not address the subject health professional; rather, it addresses knowledge and attitude, so I suggest you consider rewriting it to “Assessment of knowledge and attitudes of primary care health professionals on Meningococcal, Rotavirus, and Human Papillomavirus vaccines.”
The introduction section does not describe the critical role of family physicians and family health personnel in vaccine administration and patient education. You also need to explain how knowledge and attitudes are associated with vaccine uptake and hesitancy.
Since your study used Pearson's chi-square test, I am of the opinion that this is beyond descriptive, and some element of analytical has been used. So, consider making the study design more appropriate for the scope of the analysis conducted in the study.
There are approximately 30,000 family physicians in Turkey from how many health care centres? Are each of the health centres considered in recruiting the study particpants? "Health professionals to be interviewed" may be misleading since the study used an online survey, not interviews. Provide more explanation on how the participants were approached and invited to the study. You also need to state the exact centre where the ethics approval was sought. State if the participants consented before participating and which type of consent was used.
You need to explain in detail if the questionnaire used in this study was newly developed for this research or if it was adopted/adapted elsewhere. In addition, other details regarding the use of the questionnaire, such as whether the questionnaire was validated or if a pretest/pilot study was conducted, need to be stated.
Instead of "Descriptive data were reported in frequencies and percentages," it’s more appropriate to say descriptive statistics were used to summarize categorical variables as frequencies and percentages.
The interval for the Duration of working as a health professional (years) and Duration of working in primary care health services (years) is quite wide; differences may exist between someone with 1 year and 10 years (1-10, 11-20, >20)
Author Response
Reviewer Comments, Author Responses and Manuscript Changes
Response to Reviewer 1
Comment 1: The title “Evaluation of the knowledge and attitudes of primary care health professionals on Meningococcal, Rotavirus, and Human Papillomavirus vaccines” in this manuscript addresses an important area and will provide readers with valuable information. However, the title can be modified to convey the extent of the work that was conducted. Evaluation in this present context does not address the subject health professional; rather, it addresses knowledge and attitude, so I suggest you consider rewriting it to “Assessment of knowledge and attitudes of primary care health professionals on Meningococcal, Rotavirus, and Human Papillomavirus vaccines.”
Response: Thank you for your helpful suggestion. We revised the title accordingly. The new title is: “Primary Healthcare Professionals’ Knowledge and Attitudes Towards Meningococcal, Rotavirus, and HPV Vaccines in Children and Adolescents.” We believe this version better reflects the scope and content of the study.
Comment 2: The introduction section does not describe the critical role of family physicians and family health personnel in vaccine administration and patient education. You also need to explain how knowledge and attitudes are associated with vaccine uptake and hesitancy.
Response: We have expanded the Introduction section by emphasizing the key role of family physicians and family health staff in immunization services and patient education. Additionally, we included references to previous studies that highlight the association between healthcare professionals’ knowledge and attitudes and vaccine uptake or hesitancy.
Comment 3: Since your study used Pearson's chi-square test, I am of the opinion that this is beyond descriptive, and some element of analytical has been used. So, consider making the study design more appropriate for the scope of the analysis conducted in the study.
Response: Thank you for your valuable comment. We agree with your suggestion. We have updated the title of Table II to: “Relationship between participants' characteristics and their attitudes towards non-routine vaccines, RV, MV, and HPV.” In addition, we have clarified in the methods section that this study is not purely descriptive, but rather a cross-sectional analytical study.
Comment 4: There are approximately 30,000 family physicians in Turkey from how many health care centres? Are each of the health centres considered in recruiting the study participants? “Health professionals to be interviewed” may be misleading since the study used an online survey, not interviews. Provide more explanation on how the participants were approached and invited to the study. You also need to state the exact centre where the ethics approval was sought. State if the participants consented before participating and which type of consent was used.
Response: Not all family health centers in Turkey were contacted. Participants were recruited using voluntary response sampling from a random selection of centers located in various geographical regions. Invitations were distributed via email and social media platforms. The term “interview” has been removed. Ethical approval was obtained from Kastamonu University Ethics Committee, and electronic informed consent was obtained from all participants.
Comment 5: You need to explain in detail if the questionnaire used in this study was newly developed for this research or if it was adopted/adapted elsewhere. In addition, other details regarding the use of the questionnaire, such as whether the questionnaire was validated or if a pretest/pilot study was conducted, need to be stated.
Response: The questionnaire was developed specifically for this study based on a review of relevant literature. A pilot study was conducted with 50 healthcare professionals to assess the clarity and usability of the questions. Since this was not a standardized scale, no formal validation or reliability testing (e.g., Cronbach’s alpha) was applied. We have clarified this in the Methods section.
Comment 6: Instead of 'Descriptive data were reported in frequencies and percentages,' it’s more appropriate to say descriptive statistics were used to summarize categorical variables as frequencies and percentages.
Response: Thank you for the correction. We revised the sentence to: “Descriptive statistics were used to summarize categorical variables as frequencies and percentages.”
Comment 7: The interval for the Duration of working as a health professional (years) and Duration of working in primary care health services (years) is quite wide; differences may exist between someone with 1 year and 10 years (1-10, 11-20, >20).
Response: You are right in noting the broad categories. However, subgroup analyses showed no significant differences even within narrower intervals (e.g., 1–5, 6–10, etc.). Therefore, to maintain clarity and avoid unnecessary fragmentation, we retained the original groupings.
Reviewer 2 Report
Comments and Suggestions for Authors
healthcare-3541124
Reviewer Comments
- The Introduction is clear and well written.
- Method (page 3 of 11): More information is needed concerning the universe and recruitment from “each family health center.” Describe the process. Were all family health centers in Turkey contacted? Was there non-response or did any decline? The details about randomization withing the consenting family health centers are clear.
- Method (page 3): stated participants were interviewed, but also states they completed an online questionnaire. This is unclear and seems inconsistent.
- Method (page 3): Details concerning questionnaire development are lacking. What was the process? How were items developed? Explain Likert-type frequency scale. How is this study connected to the pilot study? What steps were taken to establish reliability and validity? Pilot testing? Samples of the 39 questions are needed. What were the open-ended questions? This lack of detail is a major shortcoming of the manuscript.
- Method (page 3): What was the approach to collecting demographic information?
- Method and Results, (page 3): How did you deal with non-response? Were participants contacted only one or multiple times? What was the response rate to come up with 700 participants? Did you oversample with replacement to reach a total of 700? Important details are missing.
- Adequacy of Results information cannot be assessed without additional information concerning the questionnaire. Information from the tables is inadequate to understand the scope of 39 questions.
- Discussion: Please consider that attitudes are only one possible predictor of vaccination behavior. The decision to vaccinate may not track attitude about a vaccine. For example, consider theories such as the theory of planned behavior which includes predictors such as norms (what other providers do) and control (am I empowered to make the vaccination decision) in addition to attitudes.
Author Response
Reviewer Comments, Author Responses and Manuscript Changes
Response to Reviewer 2
Comment 1: The Introduction is clear and well written.
Response: We appreciate your positive feedback regarding the Introduction. No changes were made to this section as it was deemed sufficiently clear.
Comment 2: Method (page 3 of 11): More information is needed concerning the universe and recruitment from “each family health center.” Describe the process. Were all family health centers in Turkey contacted? Was there non-response or did any decline? The details about randomization within the consenting family health centers are clear.
Response: Thank you for this important point. Not all family health centers in Turkey were contacted. Participants were recruited voluntarily from family health centers located in various geographical regions of the country. Invitations to participate were shared via email and social media platforms. Random selection was applied to regions, not to centers individually. These details have been added to the Methods section.
Comment 3::Method (page 3): stated participants were interviewed, but also states they completed an online questionnaire. This is unclear and seems inconsistent.
Response: You are absolutely right. The word “interview” was used in error. We have corrected all such instances throughout the manuscript to reflect that the data collection was conducted through an online questionnaire, not interviews.
Comment 4: Method (page 3): Details concerning questionnaire development are lacking. What was the process? How were items developed? Explain Likert-type frequency scale. How is this study connected to the pilot study? What steps were taken to establish reliability and validity? Pilot testing? Samples of the 39 questions are needed. What were the open-ended questions? This lack of detail is a major shortcoming of the manuscript.
Response: Thank you for this comprehensive feedback. The Methods section has been significantly revised to include the following details: - The questionnaire was developed by the authors based on a review of similar studies in the literature. - A pilot study with 50 participants was conducted to test clarity and usability. - Since this was a custom-developed instrument and not a standardized scale, it was not subjected to formal validity and reliability testing. - Several open-ended questions were included to allow participants to elaborate on their reasons for recommending or not recommending vaccines. - The full questionnaire is provided as Supplementary File 1.
Comment 5: Method (page 3): What was the approach to collecting demographic information?
Response: Demographic data were collected via the online questionnaire. Participants self-reported their age, gender, profession, years of experience, education level, and region of work. These variables were presented in Section 1 of the survey (Supplementary File). This information has been clarified in the revised Methods section.
Comment 6: Method and Results, (page 3): How did you deal with non-response? Were participants contacted only one or multiple times? What was the response rate to come up with 700 participants? Did you oversample with replacement to reach a total of 700? Important details are missing.
Response: Participants were contacted only once. No follow-ups or reminders were sent. Participation was completely voluntary. No oversampling was performed. The sample size was estimated in advance using G*Power software, and data collection was closed upon reaching the required number of participants (n = 700). These methodological details have been clarified in the revised manuscript.
Comment 7: Adequacy of Results information cannot be assessed without additional information concerning the questionnaire. Information from the tables is inadequate to understand the scope of 39 questions.
Response: We appreciate your observation. To address this, the full version of the 39-question survey has been added as Supplementary File 1, and the Results section has been expanded to better reflect the questionnaire content.
Comment 8: Discussion: Please consider that attitudes are only one possible predictor of vaccination behavior. The decision to vaccinate may not track attitude about a vaccine. For example, consider theories such as the theory of planned behavior which includes predictors such as norms (what other providers do) and control (am I empowered to make the vaccination decision) in addition to attitudes.
Response: Thank you for this valuable suggestion. We have added a paragraph to the Discussion section referencing the Theory of Planned Behavior, highlighting that attitudes are only one factor influencing behavior. Other constructs, such as subjective norms and perceived behavioral control, are also mentioned and discussed in the context of vaccine recommendation behavior, with supporting literature citations.
Reviewer 3 Report
Comments and Suggestions for Authors
Review Report
Dear Editor/Authors,
Thank you for the opportunity to review this manuscript. Here are my comments for the manuscript in response to my recommendation.
More description is needed for the questionnaire scoring system, validity and reliability.
Data collection method.
Ethical part?
What are the levels of their knowledge and attitudes i.e, low, moderate, or high?
What about the open-ended questions that you mentioned in methodology.
I have confused about how you determined the questions and scores of the knowledge and attitude.
The conclusion should be based on the findings.

NA
Author Response
Reviewer Comments, Author Responses and Manuscript Changes
Response to Reviewer 3
Comment 1: Title – It may be better to change “Primary Care Health Professionals” to “Primary Healthcare Professionals.”
Response: Thank you for your suggestion. We have revised the terminology in the title and throughout the manuscript for consistency. The term “Primary Healthcare Professionals” is now used uniformly.
Comment 2: Abstract Lines 37 and 38 – Can you determine the percentage of the knowledge and attitudes?
Response: In the Results section of the manuscript, detailed percentages are provided: “For example, the most common reason for not recommending MV (61.5%) and HPVV (72.5%) was a lack of knowledge about the vaccine and dosage, whereas for RV it was the high price (50%). However, concerns about the side effects of vaccines was among the reasons why the vaccine was not recommended.” Due to space limitations in the abstract, we included a summarized version of this information to maintain clarity and conciseness.
Comment 3: Line 39 “Health workers”, it is better to unify the terms; you used healthcare professionals. Introduction: The gap in the literature should be described. Previous studies, either nationally or internationally, should be mentioned.
Response: Thank you for highlighting this. All instances of “health workers” and “health professionals” have been standardized to “healthcare professionals” throughout the manuscript to ensure consistency. Additionally, the Introduction section has been revised to better describe the gap in the literature and reference relevant national and international studies.
Comment 4: Line 83 – “Health professionals” or “healthcare professionals” as in the Abstract, please, unify.
Response: Thank you for pointing this out. The terminology has been standardized throughout the manuscript, and “healthcare professionals” is now used consistently.
Comment 5: More description is needed for the questionnaire scoring system, validity and reliability.
Response: As the questionnaire was developed specifically for this study and was not a standardized scale, no formal scoring, validity, or reliability testing was applied. We have clarified this in the revised Methods section.
Comment 6: Is the questionnaire developed in this study or previously used?
Response: The questionnaire was designed by the authors and developed specifically for this study, based on a review of related literature. It had not been previously used in other research.
Comment 7: The data collection method should be discussed.
Response: Data collection was carried out via an online questionnaire shared through email and social media channels. We have expanded the Methods section to provide a detailed explanation of this process.
Comment 8: The ethical part should be discussed, confidentiality…. etc.
Response: Ethical approval was obtained from Kastamonu University Ethics Committee. Participation was voluntary, and informed consent was obtained electronically at the beginning of the questionnaire. Anonymity and confidentiality were ensured. These details are now included in the Methods section.
Comment 9: What are the levels of their knowledge and attitudes, i.e., low, moderate, or high?
Response: Since this was not a standardized scale, we did not assign numerical scores to categorize levels such as “low,” “moderate,” or “high.” Instead, attitude-related items were analyzed based on whether participants recommended the vaccines or not. This is explained in the revised manuscript.
Comment 10: What about the open-ended questions that you mentioned in the methodology?
Response: Open-ended questions were included to capture participants’ qualitative feedback on reasons for recommending or not recommending certain vaccines. The content and purpose of these questions are explained in the Methods section. The full questionnaire, including open-ended items, is provided as Supplementary File 1.
Comment 11: I am confused about how you determined the questions and scores of the knowledge and attitude. Can you discuss the scoring system for each part?
Response: As mentioned earlier, no formal scoring system was used because the questionnaire was not based on a validated scale. Responses were analyzed descriptively, and attitudes were assessed based on whether participants supported vaccination practices or not.
Comment 12: The conclusion should be based on the findings.
Response: We have revised the Conclusion section to more clearly reflect the main findings of the study. Emphasis has been placed on the observed gaps in knowledge and variations in recommendation patterns, with specific conclusions aligned with the data.
Round 2
Reviewer 2 Report
Comments and Suggestions for Authors
My previous comments and concern have been addressed. The manuscript is greatly improved.
Author Response
Dear Reviewer,
Thank you very much for your valuable feedback and encouraging comments regarding the revised version of our manuscript.
We are truly grateful that you found the changes satisfactory and considered the manuscript significantly improved. Your constructive input throughout the review process has played a key role in enhancing the clarity and quality of our work.
With sincere appreciation,
Dr. Eren Yıldız
On behalf of all authors
Reviewer 3 Report
Comments and Suggestions for Authors
Dear Authors, thank you for your responses. Just the questionnaire limitations should be mentioned in the limitations part.
Best regards
Comments on the Quality of English LanguageNA
Author Response
Dear Reviewer,
Thank you very much for your additional comment and for reviewing our revised manuscript once again. As suggested, we have added a clear statement regarding the limitations of the questionnaire to the Limitations section of the Discussion.
We appreciate your close reading and valuable contribution, which helped us further strengthen our manuscript.
Kind regards,
Dr. Eren Yıldız
On behalf of all authors